# Porous single-crystalline titanium dioxide at 2 cm scale delivering enhanced photoelectrochemical performance

Fangyuan Cheng[1], Guoming Lin[1], Xiuli Hu[1], Shaobo Xi[1] & Kui Xie [1]

Porous single-crystalline (P-SC) titanium dioxide in large size would significantly enhance their photoelectrochemical functionalities owing to the structural coherence and large surface area. Here we show the growth of P-SC anatase titanium dioxide on an 2 cm scale through a conceptually different lattice reconstruction strategy by direct removal of K/P from $KTiOPO_4$ lattice leaving the open Ti-O skeleton simultaneously recrystallizing into titanium dioxide. The (101) facet dominates the growth of titanium dioxide while the relative titanium densities on different parent crystal facets control the microstructures. Crystal growth in reducing atmospheres produces P-SC $Ti_nO_{2n-1}$ ($n = 7\sim38$) in magneli phases with enhanced visible-infrared light absorption and conductivity. The P-SC $Ti_nO_{2n-1}$ shows enhanced exciton lifetime and charge mobility. The P-SC $Ti_nO_{2n-1}$ boosts photoelectrochemical oxidation of benzene to phenol with P-SC $Ti_9O_{17}$ showing 60.1% benzene conversion and 99.6% phenol selectivity at room temperature which is the highest so far to the best of our knowledge.

[1] CAS Key Laboratory of Design and Assembly of Functional Nanostructures, and Fujian Provincial Key Lab of Nanomaterials, Fujian Institute of Research on the Structure of Matter, Chinese Academy of Sciences, Fuzhou, Fujian 350002, China. Correspondence and requests for materials should be addressed to K.X. (email: kxie@fjirsm.ac.cn)

Titanium dioxide has been receiving widespread attentions in solar energy conversion[1–7]. The effectiveness of energy conversion is dedicated to a great extent by the capability of semiconductor itself including effective suppression of rapid electron/hole recombination and efficient light absorption in visible–infrared region[8,9]. The suppression of charge recombination requires instantaneous charge separation, transport and collection as well as large surface area to host surface reactions. Porous single-crystalline (P-SC) $TiO_2$ would significantly enhance these functionalities owing to the unique advantage by the combination of structural coherence and large surface area[10–12]. The resolved long-range ordering features would significantly reduce the recombination center and the electron/hole scattering in these grainboundary-free $TiO_2$ skeletons.

Crystal growth is normally along fixed directions while the inside pores are typically considered as inclusion defects in bulk crystals, which makes it extremely difficult to directly grow porous single crystals using traditional approaches[13,14]. P-SC $TiO_2$ nanoparticles at ~1 μm scale have been prepared using template approaches[15,16]; however, electrode assembly would require proper loading of nanoparticles that produces contact interfaces[15]. P-SC $TiO_2$ at centimeter-scale would reduce the grain boundaries and contact interfaces in electrode assembly at the largest extent. The photoelectrochemical performance would be significantly enhanced by the combination of highly accessible surface areas and the long-range electronic connectivity. P-SC $TiO_2$ at centimeter-scale would therefore demonstrate huge potential both in fundamental research and practical applications.

The wide band gap of $TiO_2$ (~3.2 eV) considerably limits the optical absorption under sunlight. Visible–infrared light absorption has been achieved through tailoring the chemical composition by doping either metal or nonmetal in lattice which produces localized defect structures that generate donor or acceptor states in the band gap[17,18]. Self-doping with $Ti^{3+}$ in $TiO_2$, different from impurity incorporation, is another effective approach to improve visible-light absorption[19,20]. These doping strategies with point defects improve the visible-light absorption to some extent either by lowering the conduction band or by upgrading the valence band. Black $TiO_2$ through hydrogenation has also been demonstrated for the core-shell nanoparticles in which the crystalline $TiO_2$ quantum structures are covered with disordered phases[21,22]. Here we show a different approach of band gap engineering to achieve visible-infrared light absorption through tailoring the electronic structures by the incorporation of disordered $Ti^{3+}$ interstitials in magneli phases.

In a photoelectrochemical cell, the photogenerated holes on $TiO_2$ photoanode surface would readily oxidize water in electrolyte solution to evolve oxygen while the electrons transport to counter electrode and reduce proton into hydrogen under light irradiation and external bias[8,23–25]. The generation of •OH radical from $H_2O$ is an important step in this photo-oxidation process, which provides a unique opportunity of direct utilization of the highly active •OH radical to facilitate heterogeneous oxidation catalysis before the •OH radicals themselves transform into oxygen. Direct catalytic conversion of benzene to phenol is one of the most active topics in fundamental and applied research[26,27]. The highly-stable C–H bond of benzene requires a reaction temperature of 50–140 °C using efficient catalysts[28–30]. Highly active •OH radical would to facilitate the C–H bond activation and accordingly convert benzene to phenol at room temperature. Here we demonstrate highly-efficient conversion of benzene to phenol using P-SC $TiO_2$ photoanode in photoelectrochemical cells.

In this work, we demonstrate the growth of P-SC $Ti_nO_{2n−1}$ ($n$ = 7~38) at 2 cm scale and discuss the growth mechanism in relation to lattice reconstruction. We engineer the magneli phases of P-SC $Ti_nO_{2n−1}$ aiming to enhance visible-infrared light absorption. We show the enhanced performance of photoelectrochemical water splitting to oxygen and oxidation of benzene to phenol using P-SC $Ti_nO_{2n−1}$ photoanodes.

## Results

**Crystal growth.** We grow $KTiOPO_4$ (KTP) crystals and cut them into substrates (10 mm × 20 mm × 0.5 mm)[31,32]. Supplementary Fig. 1 gives the crystal structure and facet roughness of KTP substrates in which the facets are (100), (010) and (001) along the $a$-axis, $b$-axis and $c$-axis, respectively. Figure 1a shows the X-ray diffraction (XRD) patterns of P-SC anatase $TiO_2$ grown along the $a$-axis, $b$-axis and $c$-axis of the KTP substrates while the inset is the crystal structure viewed from <101> direction. The chemical formula is confirmed to be $Ti_{38}O_{75}$ by direct detection of CO generation when reducing $TiO_2$ with graphite in vacuum system[33,34]. The single-crystalline feature of anatase $Ti_{38}O_{75}$ indicates the competitive growth of <101> orientation along the $a$, $b$ and $c$ directions of KTP substrates. Figures 1b–d show the microstructure of $Ti_{38}O_{75}$ grown along the $a$-axis, $b$-axis and $c$-axis of KTP, respectively. The dimensions of porous crystals remain similar to that of parent crystals, which therefore creates porosity by removing K/P/O atoms. The microstructures with pores in the range of 50-100 nm, and the porosity of ~60% is well coincident with the calculated values. Figure 1c, d shows the microstructures of P-SC $Ti_{38}O_{75}$ crystals grown on along the $b$-axis and $c$-axis of KTP substrates, respectively. Although they demonstrate similar porosities, the microstructures with relatively separated islands are observed, which may be ascribed to the dissociation of chain structure of Ti-O octahedron and lower Ti density along the $b$-axis and $c$-axis of KTP. The porous single crystals are grown only along the <101> direction, which may be due to that the (101) facet is the low-index facet with the lowest surface free energy[35–39] as shown in Supplementary Fig. 2. And the defect formation energy is gradually decreased with smaller $n$ values in $Ti_nO_{2n−1}$ system; however, they are still much lower than zero, which indicates an exothermic process that would favor the formation of magneli phases.

We use a transmission electron microscopy (TEM) coupled with focused ion beam (FIB) to examine the nature of microstructures of the P-SC anatase $Ti_{38}O_{75}$ crystals. Figure 2 shows the cross-sectional view of porous crystal grown along the $a$-axis of KTP, which further confirms the distribution of interconnected pores with the diameter of 50–100 nm. The selected area electron diffraction (SAED) at different locations on the porous skeleton shows identical facet orientations and single-crystalline nature. We further show the microstructures and single-crystalline nature of porous crystals grown along with the $b$-axis and $c$-axis of KTP as shown in Supplementary Figs. 3 and 4. P-SC crystals not only keep the single-crystalline nature but also maintain the porous microstructures when the crystal growth performs in a stronger reducing atmosphere (Ar/$H_2$ atmospheres, 67–333 mbar at 600–800 °C). We take the P-SC $Ti_9O_{17}$ crystal grown in reducing atmosphere as an example shown in Supplementary Fig. 5. In this case, the further loss of oxygen leads to the presence of more $Ti^{3+}$ interstitials in bulk that generates magneli phase, which would tailor the electronic structures and band gap structures to better suit light absorption.

**Growth mechanism.** We use a spherical aberration corrected scanning transmission electron microscope (Cs-corrected STEM) coupled with FIB to investigate the P-SC $Ti_{38}O_{75}$ and P-SC $Ti_9O_{17}$ crystals. There is no H residual in the porous crystals as confirmed by a solid state Nuclear Magnetic Resonance (NMR) test. Figure 3a shows the high-resolution TEM of the P-SC anatase $Ti_{38}O_{75}$ grown along the $a$-axis of KTP parent crystal. The

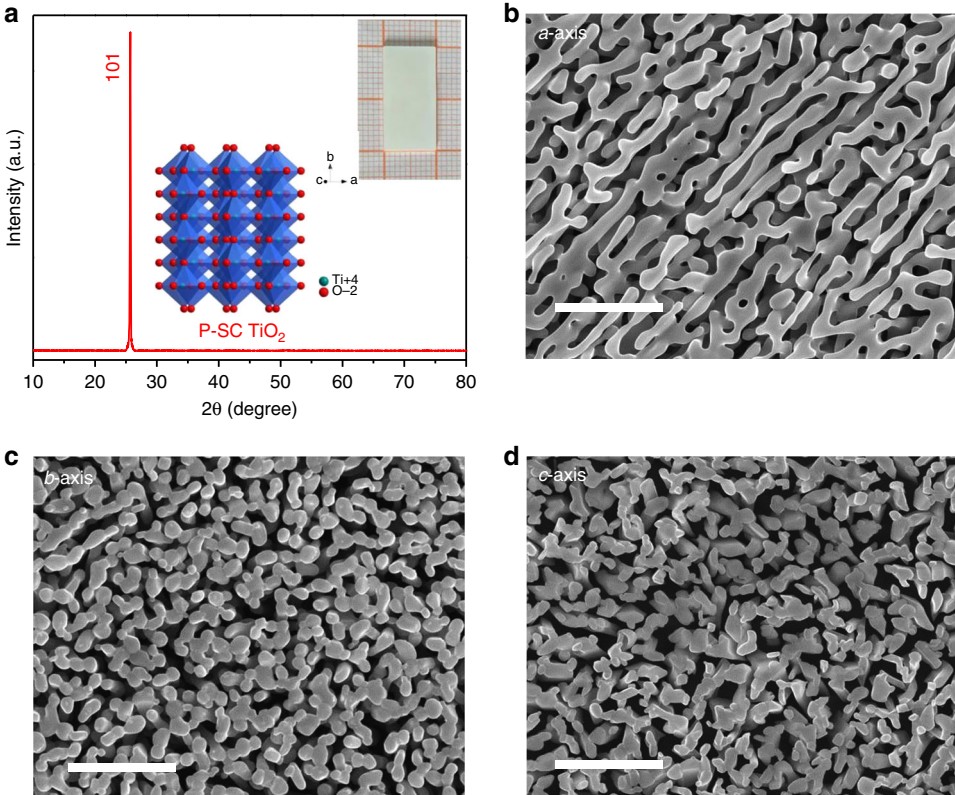

**Fig. 1** XRD and SEM characterization of porous single-crystalline (P-SC) anatase $TiO_2$ crystals. **a** the XRD of P-SC $TiO_2$ crystals grown along the *a*-axis, *b*-axis and *c*-axis of $KTiOPO_4$ (KTP) substrates. The inset is the crystal structure of anatase $TiO_2$ view along 101 axis. **b** the SEM of P-SC $Ti_{38}O_{75}$ crystals grown along the *a*-axis of KTP. **c** the SEM of P-SC $Ti_{38}O_{75}$ crystals grown along the *b*-axis of KTP. **d** the SEM of P-SC $Ti_{38}O_{75}$ crystals grown along the *c*-axis of KTP. The KTP substrates with dimensions of 20 mm × 10 mm × 0.5 mm are used for the growth of P-SC $Ti_{38}O_{75}$ crystals in Ar at 600–800 °C. The scale bar is 1 μm in **b**, **c** and **d**

lattice spacing of 0.237 and 0.352 nm could be assigned to (002) and (011) fringe as further confirmed by the SEAD pattern shown in the inset[33]. A slight content of random dislocations is present in the anatase $Ti_{38}O_{75}$ crystal while the single-crystalline features remain unchanged. Crystal growth in a stronger reducing atmosphere ($H_2$/Ar atmosphere, 67–333 mbar at 600–800 °C) produces the P-SC anatase $Ti_9O_{17}$ in magneli phase as shown in Fig. 3b. The oxygen loss produces high concentration of $Ti^{3+}$ interstitials in bulk while periodical dislocations are present to tolerate these point defects. We use high-sensitive low-energy ion scattering (HS-LEIS) with $He^+$ (3 keV) and $Ne^+$ (5 keV) ion resources to analyze the atomic surface termination layer. The $Ne^+$ ions scattering could detect surface heavy elements like Ti atom while the $He^+$ ion scattering is more sensitive to O atoms[14]. Both P-SC $Ti_{38}O_{75}$ and $Ti_9O_{17}$ crystals terminate with Ti-O skeleton on (101) facets even though there is oxygen loss in reduced crystal as shown in Fig. 3c. The atomic termination layer on (101) facet is well consistent with the stabilized structure shown in Supplementary Fig. 2c. X-ray Photoelectron Spectroscopy (XPS) results in Fig. 3d show that the Ti is mainly +4 in the P-SC $Ti_{38}O_{75}$ while the contents of $Ti^{3+}$ are accordingly increased in $Ti_nO_{2n-1}$ ($n$ = 7~38) with the decrease of n values in magneli phases. We finally obtain different P-SC anatase $Ti_nO_{2n-1}$ crystals in magneli phases by gradually varying growth atmospheres and temperatures as shown in Supplementary Fig. 6a. In Supplementary Fig. 6b–f, we observe similar microstructures for the different $Ti_nO_{2n-1}$ crystals in magneli phases though higher oxygen loss leads to more compressed microstructures.

We conduct the measurements of Raman spectroscopy of the porous $Ti_nO_{2n-1}$ single crystals as shown in Supplementary Fig. 7. The peaks at 145, 197, 395, 515, and 636 $cm^{-1}$ are well consistent with the anatase phase of $TiO_2$[40,41] through slight Raman shift is observed for different $Ti_nO_{2n-1}$ compositions. We further conduct Brunauer-Emmett-Teller (BET) tests of the porous $Ti_nO_{2n-1}$ single crystals as shown in Supplementary Fig. 8. These porous single crystals demonstrate similar surface areas (~7 $m^2 g^{-1}$) even for the crystals with different chemical compositions. And the mean pore sizes are in the range of 80-100 nm which are well consistent with the SEM results. Figure 4a shows the crystal structure of KTP viewed from the *a*-axis which demonstrates the vertical and periodical lattice channels of K ions while the P-O polyhedrons are closely distributed[42]. The removal of K ions would proceed accompanied with P-O polyhedron collapsing while the channels would facilitate the atom diffusion leaving the open $TiO_2$ skeleton in Fig. 4b. To maintain a low-energy state, the left $TiO_2$ skeleton finally transforms into porous anatase $TiO_2$ single crystals while the growth is dominated by the low energy (101) facet as shown in Fig. 4c[43]. The energy barrier of removing K atom through the lattice channel is only 3.58 eV and the removal of O linked to P in polyhedron is as low as 3.52–4.41 eV as shown in Supplementary Fig. 9. The removal of O linked to P may lead to the P-O polyhedron collapsing together with the evaporation of P atom from KTP lattice. The atomic diffusion of K/P/O atoms in lattice would finally lead to the transformation of the KTP crystal into anatase $TiO_2$ crystal in Fig. 4d.

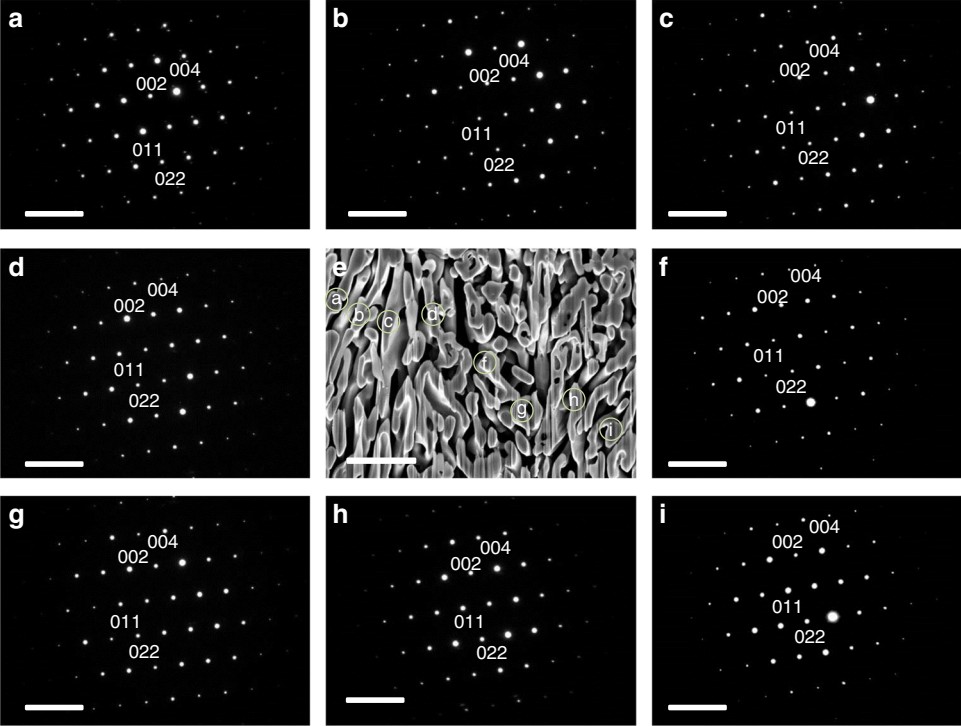

**Fig. 2** Cross-sectional view and selected area electron diffraction (SAED). The (**a–d**) and (**f–i**) present the SAED pattern at different locations on the skeleton of the P-SC anatase $Ti_{38}O_{75}$ crystal. The e presents the cross-sectional view of the P-SC anatase $Ti_{38}O_{75}$ crystal while the locations for SAED pattern are labeled. The porous single-crystalline (P-SC) anatase $Ti_{38}O_{75}$ crystal is grown along with the *a*-axis of KTP substrate. The scale bar is 5 1/nm in (**a–d**) and (**f–i**). The scale bar is 1 μm in (**e**)

**Crystal property.** Figure 5a shows the ultraviolet-visible spectroscopy of the P-SC anatase $Ti_nO_{2n-1}$ crystals ($n = 7$–38) between 200 and 800 nm. With the increase of $Ti^{3+}$ interstitial concentration, the visible-infrared light absorption gradually becomes stronger and finally demonstrate nearly complete light absorption. P-SC $Ti_nO_{2n-1}$ crystals ($n \leq 25$) with significant concentration of $Ti^{3+}$ actually become black and electronic conductors. We also consider the possible influence of the dipole moment of (101) facet in $Ti_nO_{2n-1}$ with different ratio between Ti and O atoms. As shown in Supplementary Fig. 10, the dipole moment is approximately −0.2 Debye even though slight fluctuation is observed for $Ti_nO_{2n-1}$ with different $n$ values. The dominance to enhance light absorption would be the band gap engineering through control of tailored electronic structures. We calculate the band structures for the pure and reduced $TiO_2$ (101) surface with HSE06 hybrid density functional by VASP software[44,45]. The calculated total density of states (TDOS) for pure $TiO_2$ (101)−(1 × 4) surface unit cell and reduced $TiO_2$ with Ti interstitial are summarized in Fig. 5b while the projected density of states (PDOS) is shown in Supplementary Fig. 11. For pure $TiO_2$, the Fermi level is located just above its valence band maximum, indicating the typical properties of semiconductor with band gap of 3.29 eV. The conduction band is mainly composed of Ti-3d orbitals while O-2p orbitals dominate the valence band. In contrast, the Fermi level goes through the conduction band for $Ti_9O_{17}$ with $Ti^{3+}$ interstitial in magneli phase, which narrows the band gap down to only 1.12 eV while the valance band position remain unchanged. The anatase $Ti_nO_{2n-1}$ ($n = 7$–38) in magneli phases would extend the photoabsorption to the visible-infrared range, which is well consistent with our experimental results. The free electrons filled in the bottom of conduction band would make conductivity significantly improved.

We further study the transient absorption spectroscopy of the P-SC $Ti_nO_{2n-1}$ ($n = 7$–38) under excitation as shown in Fig. 5c and Supplementary Fig. 12, which confirms the unusual lifetime of exciton (~10 ns) in P-SC $Ti_nO_{2n-1}$ ($n = 7$–38) crystals. This lifetime is comparable to that of bulk crystals and ~10 times higher than that of polycrystalline materials, indicating the considerably enhanced suppression of charge recombination with structural coherence[46,47]. The fluorescence decay in Supplementary Fig. 13 further validates the long lifetime of photoexcited charge in P-SC $Ti_nO_{2n-1}$ ($n = 7$–38) crystals. Similar $\tau_1$ values which indicate the lifetime of free electron–hole recombination in bulk are obtained while similar $\tau_2$ values which indicate the lifetime of electron–hole recombination on surface are observed, which may be due to the similar structural coherence of $Ti_nO_{2n-1}$ single crystals even with different chemical compositions. P-SC $Ti_nO_{2n-1}$ crystals with smaller n values give rise to enhanced electron density and mobility that contribute to increased electronic conduction as shown in Fig. 5d and Supplementary Fig. 14. Excessive $Ti^{3+}$ interstitials lead to the decrease of electron mobility though the conductivity is further improved. As shown in Fig. 5d, the growth of titanium dioxide along the *a*-axis of parent KTP crystal in reducing atmosphere gives rise to enhanced electron mobility with the $Ti_9O_{17}$ demonstrating the best performance. However, the $Ti_7O_{13}$ shows a decreased mobility which could be due to the increase of point defects in the form of Ti interstitials in lattice. We then have further measured the porous $Ti_nO_{2n-1}$ ($n = 7$–38) single crystals grown along the *b*-axis and *c*-axis of the parent KTP crystals to check the possible fluctuations of electron mobility related to microstructures. As shown in Supplementary Fig. 14, for a fixed chemical composition, the fluctuations of electron mobility are negligible for the porous single crystals even though they are grown along the three different axis of the parent KTP crystal.

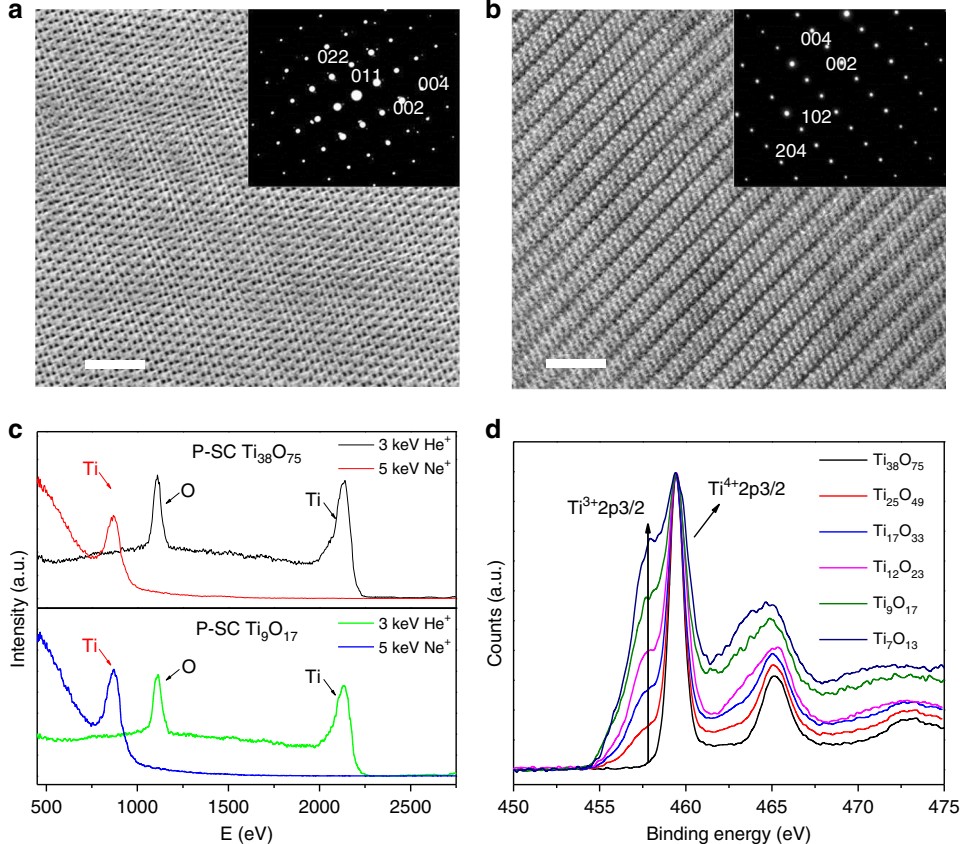

**Fig. 3** Surface and bulk structures of P-SC $Ti_nO_{2n-1}$ crystals. **a** Spherical aberration corrected Scanning Transmission Electron Microscope (Cs-corrected STEM) image of the P-SC $Ti_{38}O_{75}$ view towards (011) plane and along c axis. Inset image shows the corresponding SAED pattern of P-SC $Ti_{38}O_{75}$. **b** Cs-corrected STEM image of the P-SC $Ti_9O_{17}$ view towards (102) plane and along c axis. Inset image shows the corresponding SAED pattern of P-SC $Ti_9O_{17}$. **c** High-sensitive low-energy ion scattering (HS-LEIS) spectra of the outmost surface layer of P-SC $Ti_{38}O_{75}$ and $Ti_9O_{17}$ under the ion sources of 3 keV $He^+$ and 5 keV $Ne^+$, respectively. **d** XPS spectra of P-SC $Ti_nO_{2n-1}$ samples with different n values. The scale bar is 2 nm in (**a**) and (**b**)

**Photoelectrochemical performance.** Figure 6a shows the photocurrent-potential curves of P-SC $Ti_nO_{2n-1}$ (n = 7–25) in 1 M NaOH solution under 10 times of air mass (AM) 1.5 G irradiation. The dimensions of the free-standing P-SC $Ti_nO_{2n-1}$ single crystals are 10 mm × 20 mm × 0.5 mm with the cross-sectional view shown in the Supplementary Fig. 15a. With band gap engineering, light absorption dominates photocurrent densities in relation to the n values in the magneli phases. A volcano curve is observed with $Ti_9O_{17}$ showing the highest photocurrent density, which may be ascribed to the synergy of electronic structures and transport properties. The photocurrent density is as high as 3–9 mA cm$^{-2}$ for the P-SC $Ti_nO_{2n-1}$ (n = 7–25) photoanodes under irradiation while the dark current densities are generally below 0.5 mA cm$^{-2}$ in Supplementary Fig. 15b. Higher potentials facilitates the separation of electron and hole that further leads to enhanced photocurrent densities as shown in Supplementary Fig. 15c. The applied bias photon-to-current efficiencies (ABPEs) of $Ti_9O_{17}$ at 1.23 V is generally higher than 90% below the incident light of 400 nm, and still contribute in a similar trend with photocurrent densities in Supplementary Fig. 15d. The ABPEs with different P-SC $Ti_nO_{2n-1}$ crystals under different applied voltages ranging from 0.4 to 1.23 V are shown in Supplementary Fig. 16. It is observed that higher voltages are favorable for the enhancement of electron–hole separation which therefore leads to improved ABPEs. Although similar transient absorption spectroscopies and transient fluorescence spectroscopies are observed for the porous $Ti_nO_{2n-1}$ single crystals, we still clearly observe the gradually enhanced photocurrent densities with smaller n values.

The increase of Ti interstitial in lattice engineers and narrows the band gap to enhance the visible–infrared light absorption. In this case, the photocurrent densities would be mainly dominated by the light absorption while the transport properties of the single crystals would also deliver influences, which therefore lead to the optimum composition with the porous $Ti_9O_{17}$ single crystal showing the best performance.

Figure 6b shows the photocurrent densities versus light intensity using P-SC $Ti_9O_{17}$, nonporous single crystalline (N-SC) $Ti_9O_{17}$ and nonporous polycrystalline (N-PC) $TiO_2$ photoanodes under simulated solar light. The P-SC $Ti_9O_{17}$ shows a linear enhancement *versus* light intensity and finally reaches ~50 mA cm$^{-2}$ at 1.23 V under illumination intensity of 50 AM 1.5 G, which represents the highest photocurrents using titanium dioxide photoanode. The synergy of porous microstructure, structural coherence and transport property significantly enhances the functionalities of $TiO_2$ itself and thus contribute to this exceptionally high photoelectrochemical water oxidation performance. The N-SC $Ti_9O_{17}$ shows a saturated photocurrent densities even under illumination intensity of 20 AM 1.5 G at 1.23 V, demonstrating the limited performance with low surface area. The N-PC $TiO_2$ film (200 nm in thickness) on FTO shows the saturated photocurrent densities of ~1 mA cm$^{-2}$ under illumination intensity up to 50 AM 1.5 G, indicating the limited solar energy conversion with excess grain boundaries and interfaces in electrode[48,49]. The performance of P-SC $Ti_9O_{17}$ is ~50 times higher than that of N-PC $TiO_2$ at high light intensity. Figure 6c shows the

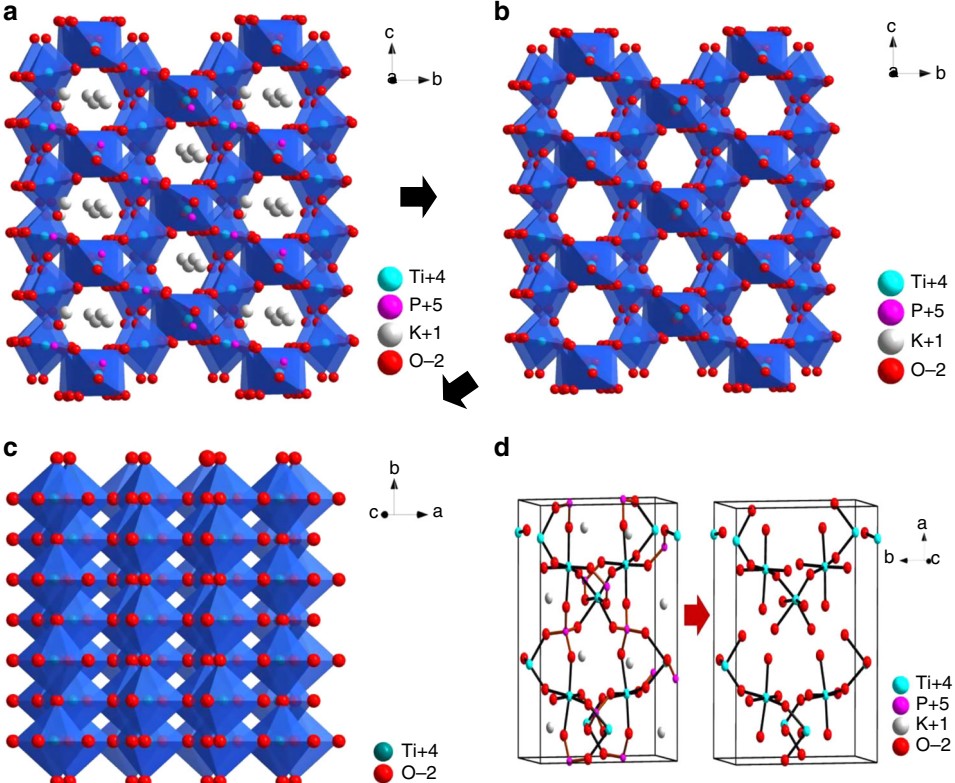

**Fig. 4** Crystal structure and lattice channel of K/P removal in KTP crystals. **a** K/P evaporation channels in a-axis KTP (view along a-axis). **b** Framework of TiO₂ by removing the K and P from the KTP (view along *a*-axis). **c** Crystal structure of anatase TiO₂ (view along 101 axis). **d** Ball and stick model of the transformation of KTP to TiO₂ (view toward 101 plane)

photocurrent density - potential curves of P-SC $Ti_nO_{2n-1}$ ($n=$ 7–25) photoanodes for the photoelectrochemical oxidation of benzene to phenol in 0.5 M $Na_2SO_4$ electrolyte under illumination intensity of 10 AM1.5 G. Similar photocurrent densities with maximum values at ~9 mA cm⁻² are observed in contrast to NaOH electrolyte while the onset potentials move forward for ~0.2 V. The dark current densities are still below 0.5 mA cm⁻² in Supplementary Fig. 17a. We operate the photoelectrochemical oxidation for a duration of 24 h in Supplementary Fig. 17b and then analyze the benzene conversion and phenol yield in Supplementary Fig. 17c and d. The generation of •OH radical would readily oxidize benzene while the large surface area in porous microstructures would host these surface reactions.

Figure 6d shows the benzene conversion and phenol yield at a constant voltage of 1.0 V using the P-SC anatase $Ti_nO_{2n-1}$ photoanodes. With band gap engineering, we observe a volcano curve for different $Ti_nO_{2n-1}$ in magneli phases with the P-SC $Ti_9O_{17}$ photoanode showing the highest benzene conversion of 60.1% and phenol selectively of 99.6%. The benzene hydroxyla-tion to produce phenol in photochemistry oxidation process is generally believed to proceed via an oxygenation pathway induced by the in situ-formed •OH radical[30,50,51]. These active •OH radicals would readily oxidize the benzene to phenol in aqueous phase. We further conduct electron spin resonance (ESR) measurement to detect the irradiated reaction system containing 5,5-dimethyl-1-pyrroline *N*-oxide (DMPO) which acts as a trapping agent of •OH radical. As shown in Supplementary Fig. 18, the observed ESR signals confirm the formation of •OH radicals during the photoelectrochemical reactions with phenol formed in this process. Negligible phenol

is formed after we add ethanol, which acts as scavenger of •OH radicals, into the reaction system. We further to detect the formation of benzene radical cation in this photoinduced process. We detect the ESR signal of benzene radical cation after we cool the aqueous solution using liquid nitrogen after 1 h reaction. We have not observed the corresponding ESR signal of benzene radical cation. Therefore, the reasonable pathway of benzene oxidation is through a •OH radical reaction in this photoelectrochemical process.

## Discussion

In conclusion, we demonstrate a conceptually different approach of lattice reconstruction strategy to grow porous titanium dioxide single crystals at on an unprecedented 2 cm scale. The synergistic control of porous microstructure, structural coherence and band gap engineering considerably enhances the functionalities of the P-SC anatase $Ti_nO_{2n-1}$ ($n=7$–38) in mageli phases. The pre-ferential growth of (101) facet dominates the growth of anatase titanium dioxide while the relative Ti densities on parent crystal facet controls the microstructures. The $Ti^{3+}$ interstitials in $Ti_nO_{2n-1}$ account for the Fermi level going through conduction band that narrows band gap down to better suit visible–infrared light absorption. We show the ultrahigh photoelectrochemical per-formance using P-SC $Ti_nO_{2n-1}$ crystals with $Ti_9O_{17}$ photoanode showing the highest benzene conversion of 60.1% and phenol selectively of 99.6%. The current work would open a new way for low-cost and high-throughput fabrication of porous single crys-tals in large scale and may be highly-adaptable as well to tailoring single-crystalline materials to enhance their functionalities in many other fields.

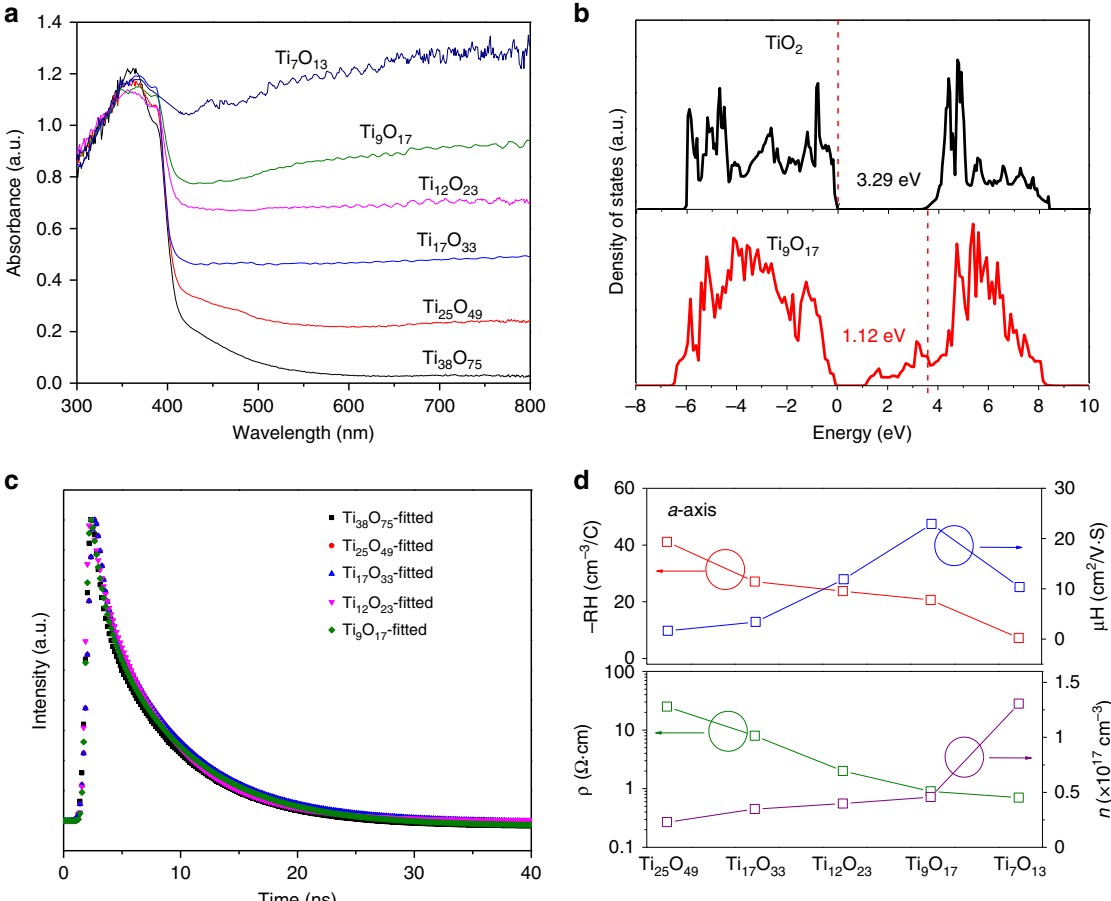

**Fig. 5** Physical properties of the porous single-crystalline (P-SC) $Ti_nO_{2n-1}$. **a** Ultraviolet-visible diffuse reflectance spectra of $Ti_nO_{2n-1}$ single crystals with different n values. **b** Density of states for $TiO_2$ and $Ti_9O_{17}$ magneli phase with $Ti^{3+}$ interstitials. The Fermi levels are shown as vertical lines. **c** Decay profiles of transient absorption of the P-SC $Ti_nO_{2n-1}$ crystals. **d** The resistivity, carrier density, Hall coefficient and Hall mobility of $Ti_nO_{2n-1}$ single crystals grown along the $a$-axis of KTP crystal substrates

## Methods

**Growth of P-SC $Ti_nO_{2n-1}$ crystals.** In this work, we firstly grow single-crystalline KTP substrates using Czochralski method and then cut them into substrates with dimensions of 10 mm × 20 mm × 0.5 mm[31,32]. The surfaces are mechanically polished while the crystal facets and roughness are analyzed using XRD on an X-ray diffractmeter (Cu-Kα, Mniflex 600) and atomic force microscopy (AFM, Bruker Dimension Edge), respectively. We then grow the P-SC $Ti_nO_{2n-1}$ crystals in vacuum system with $H_2$/Ar gas (50–200 sccm, 6 N purity) pressure controlled at 67–333 mbar at 600–800 °C. The P-SC $Ti_nO_{2n-1}$ crystals are obtained after maintaining the treatment duration for 30-60 h followed by a natural cooling process in argon gas (6 N purity).

**Characterization of microstructure and property.** We analyze the surface morphologies of P-SC $Ti_nO_{2n-1}$ samples using field-emission scanning electron microscope (FE-SEM) (Zeiss Auriga) at an accelerating voltage of 10 KV. The phase formation is then examined using XRD on an X-ray diffractmeter (Cu-Kα, Mniflex 600). We use FIB (ZeissAuriga) to prepare TEM samples and then characterize them on a Cs-TEM (FEI Titan3 G2 60–300) at 300 kV. The valence of Ti in P-SC $Ti_nO_{2n-1}$ crystals is determined using XPS on ESCALAB 250Xi. The transport properties are investigated at a physical property measurement system 9 (PPMS-9) at 300 K. We use femtosecond transient absorption spectrometer (355 nm excitation, Helios) and transient fluorescence spectrometer (375 nm excitation, FLS980) the analyze the excition dynamics under irradiation. We use the HS-LEIS spectra (ION-TOF, Qtac100) with 3 keV $He^+$ (6 nA) and 5 keV $Ne^+$ (3 nA) ion sources to analyze the atomic termination layer on porous crystals. Raman spectra of the P-SC $Ti_nO_{2n-1}$ crystals are recorded on Horiba Labram HR Evolution. Nitrogen adsorption measurements are performed at 77 K using a Micromeritics ASAP 2020C + M system utilizing Brunauer–Emmett–Teller (BET) calculations for surface area and mean pore size. Surface chemical analysis is performed by XPS (Thermal Fisher Inc., ESCALAB 250Xi). The ESR spectra were recorded on a Bruker Biospin GMBH E500 10/12 ESR spectrometer.

**Photoelectrochemical measurement.** We test the photoelectrochemical water oxidation using P-SC $Ti_nO_{2n-1}$ in aqueous 1 M NaOH on an electrochemical workstation (IM6, Zahner) with a Pt counter electrode and a saturated calomel reference electrode at 25 °C[52–54]. The test is performed in a gas-tight cell with two-compartments (50 mL) separated by an anion exchange membrane (Nafion212). The electrolyte is constantly stirred at 600 rpm to facilitate the water splitting process. We further conduct the photoelectrochemical oxidation of benzene to phenol at 25 °C in aqueous 0.5 M $Na_2SO_4$ with 20% acetonitrile to enhance benzene solubility while the benzene 0.1 ml is added in anode compartment. We use san-electric solar simulator coupled with a quartz condenser lens to provide different light intensities of standard AM 1.5 G illumination. The benzene conversion and phenol yield are determined using a gas chromatography-mass spectra on a Varian 450-GC/240-MS. The ABPE measurements are conducted in a three-electrode system with 1 M NaOH electrolyte and 350 W xenon lamp on an electrochemical workstation (IM6, Zahner, Germany). The working electrode, counter electrode and reference electrode are P-SC $Ti_nO_{2n-1}$, Pt piece and saturated calomel electrode, respectively. The external bias voltage is 1.23, 1.0, 0.8, 0.6, and 0.4 V versus RHE, respectively. All potentials are converted to RHE reference scale using the Nernst equation: $E_{RHE} = E_{Hg/Hg2Cl2} + 0.0591 \times PH + 0.244$.

**Theoretical calculation.** To understand the energy barrier of atom removal through the lattice channel and the formation of $Ti_nO_{2n-1}$ magneli phases, we calculate the defect formation energy ($E_{for}$) for the K atom, different O and Ti atoms with the following formula: $E_{for} = E_{total}$ (defect) $- E_{total}$ (perfect) $- \sum_i \Delta n_i u_i$,

where $E_{total}$ (defect) and $E_{total}$ (perfect) are the total energy of defect and perfect system, respectively. $\Delta n_i$ and $u_i$ are the the number of increase or decrease atoms and the chemical potential of the constituent atoms, respectively[55,56]. The defect configurations of the KTP (100) surface with the defect formation energies ($E_{for}$) is shown in Supplementary Fig. 9. We further calculate the band structures both the pure and reduced $TiO_2$ (101) with the HSE06 hybrid density functional using VASP software[44,45]. In our DFT calculation, the plane-wave cutoff energy of 500

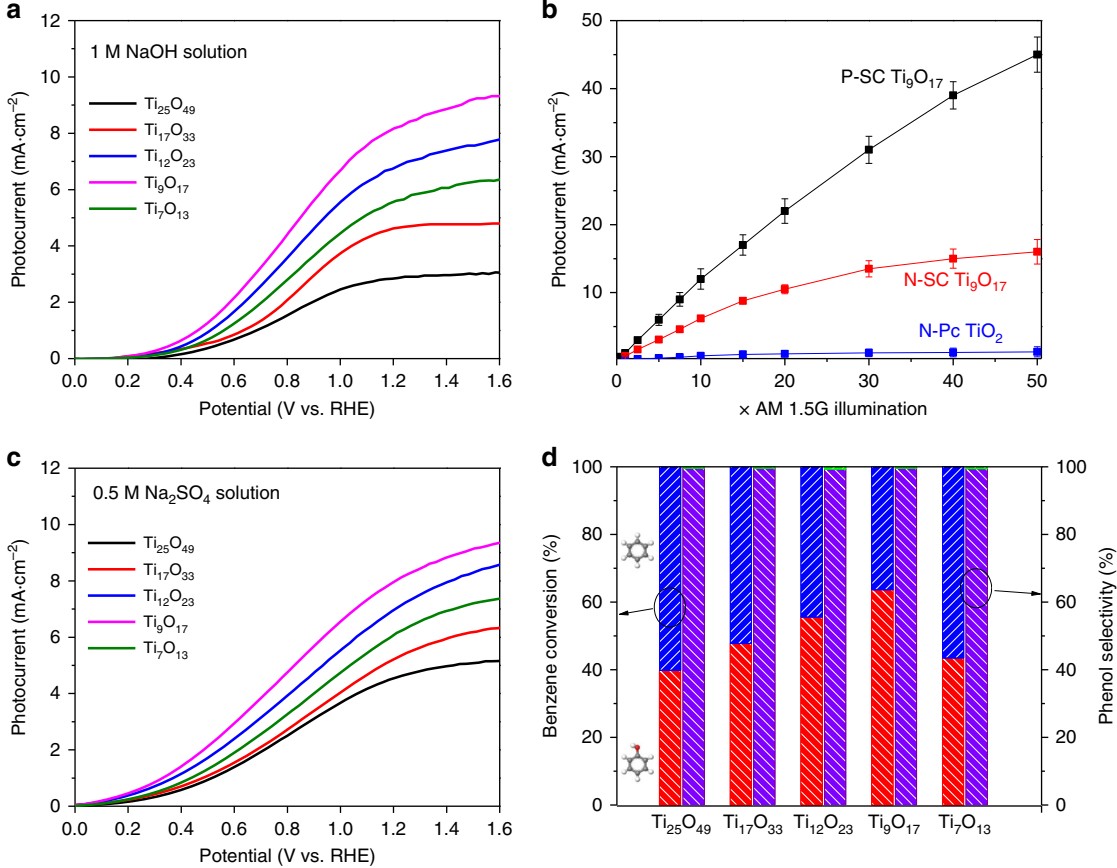

**Fig. 6** Photoelectrochemical performance of porous single-crystalline (P-SC) Ti$_n$O$_{2n-1}$. **a** Linear Scanning Voltammetry (LSV) of P-SC Ti$_n$O$_{2n-1}$ photoanodes for water oxidation using a three electrode setup (Ti$_n$O$_{2n-1}$ working, Pt counter, Hg/Hg$_2$Cl$_2$ reference electrode, scan rate of 20 mv s$^{-1}$) in a 1 M NaOH electrolyte (PH = 13.6). **b** Enhancement of photocurrent with increasing light intensity up to 50 AM1.5 G sunlight using P-SC Ti$_9$O$_{17}$, nonporous single crystalline (N-SC) Ti$_9$O$_{17}$ and nonporous polycrystalline (N-PC) TiO$_2$ electrodes harvested at 1.23 V applied bias. Error bars represent standard deviation in repeated measurements. **c** LSV of P-SC Ti$_n$O$_{2n-1}$ photoanodes for the oxidation of benzene under 10 AM1.5 G illumination. **d** Benzene conversion and phenol yield of using Ti$_n$O$_{2n-1}$ P-SC Ti$_n$O$_{2n-1}$ photoanodes

eV is used, and the energies and residual forces are converged to $10^{-6}$ eV and 0.02 eV Å$^{-1}$ in the process of electronic and geometric optimizations. We calculate the unit cell of anatase TiO$_2$ and obtained the lattice parameters: $a = b = 3.823$ Å and $c = 9.683$ Å (with a $8 \times 8 \times 3$ k-point grid) which is consistent with earlier reports[35]. As shown in Supplementary Fig. 10, the TiO$_2$ (101)−(1 × 4) surface unit cell with a wide vacuum layer (10.410 × 15.295 × 25.474 Å) is chosen to calculate the density of states for TiO$_2$ and one to four Ti atoms are intermingled to simulate Ti$_{25}$O$_{49}$, Ti$_{12}$O$_{23}$, Ti$_9$O$_{17}$ and Ti$_7$O$_{13}$ and the dipole moment of them are calculated along the z direction, respectively. K-point grids $3 \times 2 \times 1$ are adopted in the irreducible Brillouin zone for this superstructure model.

## Data availability

All reported data are included in the manuscript and supplementary materials. And the Source Data can be downloaded from the Source File: https://yunpan.360.cn/surl_yLXSUrWQP3s (Code: 3f1a)

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

## Acknowledgements

We acknowledge the funding support from the Natural Science Foundation of China (91845202, 21750110433), Dalian National Laboratory for Clean Energy (DNL180404) and Strategic Priority Research Program of Chinese Academy of Sciences (XDB2000000).

## Author contributions

F.C. and G.L. contributed equally to this work. F.C. conducted the growth of KTP single crystals, the growth of porous single crystals, microstructure characterization and photoelectrochemical measurements. G.L. conducted surface sate analysis, transport property characterization, crystal structure characterization and growth mechanism analysis. X.H. conducted the theory calculation. S.X. conduced the maintenance of vacuum system during crystal growth. All authors were involved in the data analysis and discussion. K.X. supervised this work.

## Additional information

**Competing interests:** The authors declare no competing interests.

