## [Peer Review File · Nature Communications]

Reviewers' comments:

Reviewer #1 (Remarks to the Author):

This work is to illuminate the crystal facet engineering and band gap engineering by doping with different molar ratio of Ti to Oxygen to have the highest efficiency of photodegradation and water splitting ability. The manuscript has been well organized and described to explain the results and match with photocatalytic ability and photochemical ability with crystal facet and band gap engineering. Unfortunately, this manuscript does not include clear explanations and evidences for the corresponding ones to explain it. Authors should modify and give more explanations on it by giving more clear evidences for it as followings. So it should be rejected as the present form.

1. Photon absorption ability of the different crystal facet should be based on the different dipole moment of the crystal facets. With different molar ratio of Ti and O, the photon absorption ability should be characterized and clarified based on the surface dipole moment with different molar ratio and different crystal facet.

2. To have high efficiency of photocatalysts, the electron/hole diffusivity should be characterized based on the electron/hole mobility measurements with different crystal facet and different molar ratio of Ti and O.

3. The suppression of charge recombination is also very important for the efficiency of photocatalysts. In this study, transient absorption spectroscopy of the TiO_{2n-1} P-SC was measured and shown. This shows the life-time of excited electrons. To illuminate the charge separation ability, authors also should suggest photocurrent density to illuminate the charge collection efficiency of it. These two data are not consistent each other. How it can be explained?

4. To add more clear explanations on the different decay curves and photocurrent values with different Ti to O molar ratio. It can be solved by the surface structures illumination.

All of the requirements should be explained and clarified with more clear evidences.

Reviewer #2 (Remarks to the Author):

In this work by Xie and coworkers, the authors report the fabrication of porous single-crystalline (P-SC) TiO_{2n-1} photoelectrodes with large dimensions. Such large scale single crystals of TiO_x are reported for the first time in my knowledge. The obtained P-SC TiO_{2n-1} single-crystals show high photocurrent densities toward solar water splitting and outstanding photocatalytic activities for the oxidation of benzene to phenol. The photocurrents are as high as 3-9 mA cm^{-2} for P-SC TiO_{2n-1} photoanodes under irradiation. Moreover, the TiO_{2n-1} photoanode exhibits high benzene conversion of 60.1% to phenol with high selectivity of 99.6%. The good performance is ascribed to the enhanced charge transfer due to the structural coherence, fast reactions owing to the sufficient space in the microstructures and the promoted light absorption as a result of the formation of magneli phases. The work shows high novelty in materials preparation and targeted applications in catalysis. The results show the potential to inspire further innovative works on the fabrication of porous single crystals in large scale for energy and environmental applications. I would like to recommend the publication of the manuscript in Nature Communications. Several minor revisions are suggested.

1. The authors had studied the performance of the photoelectrodes with different thicknesses. How the thicknesses of the TiO_{2n-1} films are controlled and characterized (e.g., cross-section FESEM images) should better be presented.

2. Applied bias photon-to-current efficiencies (ABPEs) of different photoelectrodes toward solar water splitting should better be provided in the manuscript.

3. In Figure 6b, the authors may need to specify the bias at which the photocurrent densities are obtained.

4. How the IPCEs are characterized should be described in the experimental section.

5. Some important references should better be cited to enrich the background of the manuscript (e.g., Teera Butburee, et al., 2D Porous TiO_2 Single-Crystalline Nanostructure Demonstrating High Photo-Electrochemical Water Splitting Performance, *Adv. Mater.*, 2018, 30, 1705666; P. Zhang, et al., Facile Synthesis of Multi-Shelled ZnS-CdS Cages with Enhanced Photoelectrochemical Performance for Solar Energy Conversion, *Chem*, 2018, 4, 162; Yong Liu, et al., Radially Oriented Mesoporous TiO_2 Microspheres with Single-Crystal-Like Anatase Walls for High-Efficiency

Optoelectronic Devices, Sci. Adv., 2015, 1, e1500166).

The manuscript is overall well prepared. Some minor changes might be needed:

- 1) The Raman spectra of the P-SC TiO_2 _{n-1} crystals should be provided.
- 2) The BET test of the P-SC TiO_2 _{n-1} crystals by should be provided.
- 3) Line 58 "the facets of 100, 010 and 001" should be "the facets of [100], [010] and [001]".
- 4) Line 61 "101 direction" should be "[101] direction".
- 5) Line 81 "suit" should be "sun".
- 6) Line 103, 111,112 and 161 "(101)" should be "[101]".
- 7) Please pay attention to some small mistakes. For example, in the caption of Figure S1, "(a, d, and g)" has been used three times. The numbering of Figure S2-S4 is confusing.

Reviewer #3 (Remarks to the Author):

This manuscript reports on the growing and characterization of porous single-crystalline anatase TiO_2 _{n-1} (n = 7-38) large size crystals and their application for photoelectrochemical oxidation of benzene to phenol which showed good benzene conversion (60.1%) and excellent phenol selectively (99.6%). The material was well characterized by variety of instrumentations and the manuscript is generally well written and clearly presented. Therefore, I recommend the manuscript to be published after minor revisions based on comments below:

- 1- It is very difficult to understand the higher selectivity towards phenol formation since phenol is more reactive than benzene and the catalytic test was carried out for 24 h. Moreover, it is well known that the photo-generated OH radical generally shows poor selectivity. Therefore, reasonable explanations should be given for the high selectivity of P-SC TiO_2 _{n-1} crystal toward phenol formation.
- 2- The authors mentioned that the photo-generated OH radical is responsible for benzene oxidation (without any evidence). However, reasonable pathway is not clear i.e. is it through direct oxidation of H₂O and formation of OH radical or indirectly via in situ formation of H₂O₂?
- 3- Additionally, phenol formation could be also initiated by photoinduced electron transfer from benzene and formation of benzene radical cation, which reacts with H₂O to yield OH-benzene adduct radical. Therefore, more detailed knowledge about the mechanism of photoelectrochemical benzene hydroxylation to phenol is required.
- 4- The high performance of P-SC TiO_2 _{n-1} is attributed to the presence of Ti(III) interstitials which enhance visible-infrared light absorption. Why then the catalytic activity of P-SC Ti₇O₁₃ is less than P-SC Ti₉O₁₇?
- 5- In Figure 6a, the authors mentioned that the Photoelectrochemical reactions were carried out in NaOH while in the manuscript (page 4) as well as in Figure 8 (SI) in KOH!
- 6- Please delete one of from Figure 9 "(b) Durability test of of"

Response to reviewers

Reviewers' comments:

Reviewer #1 (Remarks to the Author):

This work is to illuminate the crystal facet engineering and band gap engineering by doping with different molar ratio of Ti to Oxygen to have the highest efficiency of photodegradation and water splitting ability. The manuscript has been well organized and described to explain the results and match with photocatalytic ability and photochemical ability with crystal facet and band gap engineering. Unfortunately, this manuscript does not include clear explanations and evidences for the corresponding ones to explain it. Authors should modify and give more explanations on it by giving more clear evidences for it as followings. So it should be rejected as the present form.

Answer: Thank you very much for your kind comments. We respectfully do not fully agree with the referee on the research focus of our manuscript but we are willing to accept the referee views to further improve our work. We further conduct supplementary experiments and calculations to reinforce the explanations in revision.

In our work, we report the growth of porous anatase Ti_nO_{2n-1} single crystals in magneli phases at 2 cm scale using a conceptually different lattice reconstruction strategy. The (101) facet dominates the growth of TiO_2 single crystals while the crystal growth in reducing atmospheres produces Ti_nO_{2n-1} ($n = 7\sim 38$) in magneli phases. We engineer the electronic structures through control of Ti interstitials in lattice and achieve the enhanced visible-infrared light absorption. The combination of structural coherence and porosity in Ti_nO_{2n-1} single crystals shows unusual lifetime of exciton and charge mobility. The free-standing porous Ti_nO_{2n-1} single crystals deliver ultrahigh performance of photoelectrochemical oxidation of benzene to phenol with Ti_9O_{17} showing the highest performance of 60.1% benzene conversion and 99.6% phenol selectivity at room temperature. We believe the current work would open a new pathway for low-cost and high-throughput fabrication of porous single crystals in large scale and may be highly-adaptable as well to tailoring single-crystalline materials to enhance their functionalities in many other fields.

1. Photon absorption ability of the different crystal facet should be based on the different dipole moment of the crystal facets. With different molar ratio of Ti and O, the photon absorption ability should be characterized and clarified based on the surface dipole moment with different molar ratio and different crystal facet.

Answer: Thank you very much. The growth of low-energy facets would be preferential during the growth of porous anatase TiO_2 single crystals using the lattice reconstruction strategy in vacuum atmosphere at high temperatures. In our work, the porous single crystals are grown only along the <101> direction, which may be due to that the (101) facet is the low-index facet with the lowest surface free energy. As shown in Supplementary Figure 2, we summarize the relationship between surface energy and the facets of anatase TiO_2 from references, which confirms the preference of growth of (101) facet as validated in our work.^[1-2] Supplementary Figure 2(d) shows the total energy of TiO_2 (101) surface with different atomic terminations, which indicates that the lowest energy of (101) facet in TiO_2 crystal. And the stabilized structure is shown in Supplementary Figure 2(c) as reported in published work.^[3-5]

We further consider the different configurations with Ti interstitial in lattice which could give difference in the total energy. We take the $Ti_{25}O_{49}$ with exposed (101) facet as an example and calculate the total energy of different configurations with one Ti atom interstitial. As shown in Supplementary Figure 2(f), only slight fluctuation is observed for the total energy of $Ti_{25}O_{49}$ (101) even with different interstitial positions in lattice. We further calculate the defect formation energy of Ti_nO_{2n-1} system with exposed (101) facets as shown in Supplementary Figure 2(g). It is observed that the formation of Ti_nO_{2n-1} magneli phases with $n=7-38$ definitely leads to the decrease of the defect formation energy. The defect formation energy of Ti_nO_{2n-1} with (101) facet in magneli phases would therefore be mainly dominated by the concentration of Ti interstitials. However, these energies are still much lower than zero. This indicates that the formation of Ti_nO_{2n-1} magneli phases with exposed (101) facet is an exothermic process that would favor the formation of magneli phases.

Supplementary Figure 2. The energy and structures of Ti_nO_{2n-1} system. (a) The surface free energy of different crystal facets; (b-c) The optimized structure of anatase TiO_2 and the TiO_2 (101) surface, O atoms in red, Ti atoms in silvery; (d) The total energy of different termination of the TiO_2 (101) surface. A-F are corresponds to the figure above; (e) The structure of anatase TiO_2 (101) surface with one interstitial Ti atom marked in purple to simulated $Ti_{25}O_{49}$; (f) The total energy of different position of interstitial Ti atom on the TiO_2 (101) surface. I-IV corresponds to the four interstitial positions in lattice; (g) The defect formation energy of Ti_nO_{2n-1} system.

We fully agree with the referee on the relationship between light absorption ability and dipole moment of crystal facets. For a fixed chemical composition, the facets of anatase TiO_2 with higher dipole moment would be favorable for the enhancement of light absorption. In our work, only

low-index (101) facet is grown in the porous anatase Ti_nO_{2n-1} ($n = 7\sim 38$) in magneli phases at 2 cm scale. The TiO_2 (101) is nonpolar and the fluctuation of dipole moment of (101) facet in Ti_nO_{2n-1} single crystals would therefore be related to the different molar ratio between Ti and O. We further calculate the dipole moment of (101) facet in relation to the n values in anatase Ti_nO_{2n-1} single crystals. The dipole moment is ~ 0.2 Debye for (101) facet in anatase Ti_nO_{2n-1} ($n = 7\sim 25$) even though slight fluctuation of dipole moment is observed for Ti_nO_{2n-1} with different n values as shown in Supplementary Figure 10a-10i. In our work, the enhancement of light absorption ability by dipole moment would be similar for the Ti_nO_{2n-1} (101) even though the chemical compositions are different with different n values.

Supplementary Figure 10. The structures and charge density difference of Ti_nO_{2n-1} (101) system. (a-d) The optimized structures of Ti_nO_{2n-1} (101); O atoms in red, Ti atoms in silvery and the interstitial Ti atoms marked in purple for clear; (e-f) the charge density difference of Ti_nO_{2n-1} (101); The accumulation and loss of charge are represented by yellow and blue regions, respectively; (i-j) the electronic charge density difference of Ti_nO_{2n-1} (101); the dipole moment of Ti_nO_{2n-1} (101) along the z direct listed on the plots.

In our work, the band gap engineering with Ti interstitials of anatase Ti_nO_{2n-1} ($n=7\sim 38$) in magneli phases would be dominant for the light absorption. We engineer the electronic structures through control of Ti interstitials in lattice and achieve the gradually enhanced light absorption in the visible-infrared region. The free-standing porous Ti_nO_{2n-1} single crystals when used as photoanode therefore deliver ultrahigh photoelectrochemical performances.

[1] Lazzeri, M., Vittadini, A., Selloni, A. Structure and energetics of stoichiometric TiO_2 anatase surfaces. *Phys. Rev. B.* **65**, 119901 (2002).

[2] Yang, H. G., et al. Anatase TiO_2 single crystals with a large percentage of reactive facets. *Nature* **453**, 638-634 (2008).

[3] Yang, Y., Feng, Q., Wang, W., Wang, Y. First-principle study on the electronic and optical properties of the anatase TiO_2 (101) surface. *Journal of Semiconductors.* **34**, 073004 (2013).

[4] Yu, J., Low, J., Xiao, W., Zhou, P., Jaroniec, M. Enhanced Photocatalytic CO₂-Reduction Activity of Anatase TiO₂ by Coexposed {001} and {101} Facets. *J. Am. Chem. Soc.* **136**, 8839-8842 (2014).

[5] Pan, J., Liu, G., Lu, G. Q., Cheng, H. M. On the True Photoreactivity Order of {001}, {101}, and {101} Facets of Anatase TiO₂ Crystals. *Angew. Chem. Int. Ed.* **50**, 2133-2137 (2011).

2. To have high efficiency of photocatalysis, the electron/hole diffusivity should be characterized based on the electron/hole mobility measurements with different crystal facet and different molar ratio of Ti and O.

Answer: Thank you very much for your comments. Yes, the electron/hole mobility is pretty important and it may be related to the chemical compositions and crystal facets. In our work, the porous single crystalline Ti_nO_{2n-1} (n=7-38) samples are n-type semiconductors and we therefore have further measured the electron mobility for different samples. As shown in Figure 5(d), it is observed that the growth of titanium dioxide along the a-axis of parent crystal in reducing atmosphere gives rise to enhanced electron mobility with the Ti₉O₁₇, demonstrating the best performance. However, the Ti₇O₁₃ shows a decreased mobility which could be due to the increase of point defects in the form of Ti interstitials in lattice. Furthermore, only (101) facet of Ti_nO_{2n-1} (n=7-38) is grown with different parent crystal facets but slight view difference is observed in the microstructure of Ti_nO_{2n-1} crystals. We then have further measured the porous Ti_nO_{2n-1} (n=7-38) single crystals grown along the b-axis and c-axis of the parent crystals to check the possible fluctuations of electron mobility related to microstructures. As shown in Figure 5(d) and Supplementary Figure 14, for a fixed chemical composition, the fluctuations of electron mobility are negligible for the porous single crystals even though they are grown along the three different axis of the parent crystal.

Supplementary Figure 14. (a) The resistivity, carrier density, Hall coefficient and Hall mobility of Ti_nO_{2n-1} single crystals growth along the b-axis of KTP substrates. (b) The resistivity, carrier density, Hall coefficient and Hall mobility of Ti_nO_{2n-1} single crystals grown along the c-axis of KTP substrates.

3. The suppression of charge recombination is also very important for the efficiency of photocatalysis. In this study, transient absorption spectroscopy of the P-SC Ti_nO_{2n-1} was measured and shown. This shows the life-time of excited electrons. To illuminate the charge separation ability, authors also should suggest photocurrent density to illuminate the charge collection

efficiency of it. These two data are not consistent each other. How it can be explained?

Answer: Thank you very much. The lifetime of exciton of porous single crystalline Ti_nO_{2n-1} would be a reflection of the advantage of structural coherence and it can be measured using transient absorption spectroscopy. As shown in Figure 5(c) and Supplementary Figure 12, similar life time is observed for the porous single crystals, which further indicates the advantage of structural coherence of the different Ti_nO_{2n-1} ($n=7-38$) crystals. However, the band gap engineering through control of Ti interstitial in bulk dominates the light absorption. Higher concentration of Ti interstitial in bulk actually favors the significantly enhanced light absorption in the visible-infrared region, which therefore dominates the photocurrent density under irradiation.

Supplementary Figure 12. Decay profiles of transient absorption in P-SC (a) $Ti_{38}O_{75}$, (b) $Ti_{25}O_{49}$, (c) $Ti_{17}O_{33}$, (d) $Ti_{12}O_{23}$ and (e) Ti_9O_{17} .

4. To add more clear explanations on the different decay curves and photocurrent values with different Ti to O molar ratio. It can be solved by the surface structures illumination. All of the requirements should be explained and clarified with more clear evidences.

Answer: Thank you very much. We have further fitted the fluorescence decay in Supplementary

Figure 13. We have summarized the lifetime of free electron-hole recombination in bulk (τ_1) and the lifetime of electron-hole recombination on defected surface (τ_2). It is observed that the τ_1 values are similar to each other for different $\text{Ti}_n\text{O}_{2n-1}$ single crystals, which may be due to the similar structural coherence of $\text{Ti}_n\text{O}_{2n-1}$ single crystals. And similar τ_2 values are also observed for $\text{Ti}_n\text{O}_{2n-1}$ single crystals which could be due to the similar structural coherence even with different chemical compositions.

Although similar transient fluorescence spectroscopies are observed for the porous $\text{Ti}_n\text{O}_{2n-1}$ single crystals, we still clearly observe the enhanced photocurrents with smaller n values. And the porous Ti_9O_{17} single crystal shows the best photocurrent performance. The decrease of n values would lead to the increase of Ti interstitial in lattice, which accordingly engineers and narrows the band gap to enhance the visible-infrared light absorption. In this case, the photocurrent densities would be mainly dominated by the light absorption of porous $\text{Ti}_n\text{O}_{2n-1}$ single crystals while the transport properties of the single crystals would also deliver influences. The best performance with the optimum chemical composition of Ti_9O_{17} single crystal is therefore observed.

Supplementary Figure 13. The fluorescence decay curves of P-SC (a) $\text{Ti}_{38}\text{O}_{75}$, (b) $\text{Ti}_{25}\text{O}_{49}$, (c) $\text{Ti}_{17}\text{O}_{33}$, (d) $\text{Ti}_{12}\text{O}_{23}$ and (e) Ti_9O_{17} .

Reviewer #2 (Remarks to the Author):

In this work by Xie and coworkers, the authors report the fabrication of porous single-crystalline (P-SC) $\text{Ti}_n\text{O}_{2n-1}$ photoelectrodes with large dimensions. Such large scale single crystals of TiO_x are reported for the first time in my knowledge. The obtained P-SC $\text{Ti}_n\text{O}_{2n-1}$ single-crystals show high photocurrent densities toward solar water splitting and outstanding photocatalytic activities for the oxidation of benzene to phenol. The photocurrents are as high as $3\text{-}9\text{ mA cm}^{-2}$ for P-SC $\text{Ti}_n\text{O}_{2n-1}$ photoanodes under irradiation. Moreover, the Ti_9O_{17} photoanode exhibits high benzene conversion of 60.1% to phenol with high selectivity of 99.6%. The good performance is ascribed to the enhanced charge transfer due to the structural coherence, fast reactions owing to the sufficient space in the microstructures and the promoted light absorption as a result of the formation of magneli phases. The work shows high novelty in materials preparation and targeted applications in catalysis. The results show the potential to inspire further innovative works on the fabrication of porous single crystals in large scale for energy and environmental applications. I would like to recommend the publication of the manuscript in Nature Communications. Several minor revisions are suggested.

Answer: Thanks for your comments. We have further conducted supplementary experiments and fully revised our manuscript according to your comments.

1. The authors had studied the performance of the photoelectrodes with different thicknesses. How the thicknesses of the $\text{Ti}_n\text{O}_{2n-1}$ films are controlled and characterized (e.g., cross-section FESEM images) should better be presented.

Answer: Thanks for your comment and question. The dimensions of the porous $\text{Ti}_n\text{O}_{2n-1}$ single crystals are $10\text{ mm} \times 20\text{ mm} \times 0.5\text{ mm}$ in our work and these samples are free-standing. The thickness is 0.5 mm as shown in the cross-sectional view in the Supplementary Figure 15(a).

Supplementary Figure 15. (a) The cross-sectional view of free-standing P- Ti_7O_{19} single crystal electrode. (b) The dark current curves of the P-SC $\text{Ti}_n\text{O}_{2n-1}$ ($n = 7\text{-}25$) photoanodes in 1M NaOH electrolyte solution. (c) Durability test of the P-SC Ti_9O_{17} at different bias. (d) The IPCE curves of the P-SC $\text{Ti}_n\text{O}_{2n-1}$ ($n = 7\text{-}25$) photoanodes in 1M NaOH electrolyte solution.

2. Applied bias photon-to-current efficiencies (ABPEs) of different photoelectrodes toward solar water splitting should better be provided in the manuscript.

Answer: Thank you very much. We have provided the details of photocurrent measurements. In revision, we have further conducted supplementary experiments of ABPEs tests under different applied voltages ranging from 0.4 to 1.23 V as shown in Supplementary Figure 16. It is observed that higher voltages are favorable for the enhancement of electron-hole separation which therefore leads to improved IPCE efficiencies.

Supplementary Figure 16. The ABPEs curves of the porous single crystalline Ti_nO_{2n-1} ($n = 7-25$) photoanodes in 1M NaOH electrolyte solution at different bias.

3. In Figure 6b, the authors may need to specify the bias at which the photocurrent densities are obtained.

Answer: Thank you very much. We have added the details in revision. The voltage is 1.23 V for the photocurrent density measurements.

4. How the IPCEs are characterized should be described in the experimental section.

Answer: Thank you very much. We have added the details in revision. The IPCE measurements are conducted in a three-electrode system with 1M NaOH electrolyte and 350W xenon lamp on an

electrochemical workstation (IM6, Zahner, Germany). The working electrode, counter electrode and reference electrode are P-SC $\text{Ti}_n\text{O}_{2n-1}$, Pt piece and saturated calomel electrode, respectively. The external bias voltage is 1.23, 1.0, 0.8, 0.6 and 0.4 V versus RHE. All potentials are converted to RHE reference scale using the Nernst equation: $E_{\text{RHE}} = E_{\text{Hg}/\text{Hg}_2\text{Cl}_2} + 0.0591 \times \text{pH} + 0.244$.

5. Some important references should better be cited to enrich the background of the manuscript (e.g., Teera Butburee, et al., 2D Porous TiO_2 Single-Crystalline Nanostructure Demonstrating High Photo-Electrochemical Water Splitting Performance, *Adv. Mater.*, 2018, 30, 1705666; P. Zhang, et al., Facile Synthesis of Multi-Shelled ZnS-CdS Cages with Enhanced Photoelectrochemical Performance for Solar Energy Conversion, *Chem*, 2018, 4, 162; Yong Liu, et al., Radially Oriented Mesoporous TiO_2 Microspheres with Single-Crystal-Like Anatase Walls for High-Efficiency Optoelectronic Devices, *Sci. Adv.*, 2015, 1, e1500166).

Answer: Thank you very much. Yes, these references are good to enrich the background. We have cited the references in revision.

The manuscript is overall well prepared. Some minor changes might be needed:

1) The Raman spectra of the P-SC $\text{Ti}_n\text{O}_{2n-1}$ crystals should be provided.

Answer: Thank you very much. We have further conducted the measurements of Raman Spectroscopy of the porous $\text{Ti}_n\text{O}_{2n-1}$ single crystals as shown in Supplementary Figure 7. The peak at 128 cm^{-1} , 395 cm^{-1} , 515 cm^{-1} and 636 cm^{-1} are well consistent with the anatase phase of TiO_2 though slight Raman shifts are observed for different chemical compositions.

Supplementary Figure 7. Raman spectra of porous single-crystalline $\text{Ti}_n\text{O}_{2n-1}$ crystals with excitation lines at 532 nm.

2) The BET test of the P-SC $\text{Ti}_n\text{O}_{2n-1}$ crystals by should be provided.

Answer: Thank you very much. We have further conducted BET tests of the porous Ti_nO_{2n-1} single crystals as shown in Supplementary Figure 8. These porous single crystals demonstrate similar surface areas ($\sim 7 \text{ m}^2 \text{ g}^{-1}$) even for the crystals with different chemical compositions. And the mean pore sizes are in the range of 80-100 nm which are well consistent with the SEM results.

Supplementary Figure 8. (a). BET surface area of P-SC Ti_nO_{2n-1} crystals. (b). BET mean pore size of porous single-crystalline Ti_nO_{2n-1} crystals.

3) Line 58 “the facets of 100, 010 and 001” should be “the facets of [100], [010] and [001]”.

Answer: Thank you very much. We have corrected these errors in revision. The standard facet description is (100), (010) and (001).

4) Line 61 “101 direction” should be “[101] direction”.

Answer: Thank you very much. We have corrected “101 direction” into “<101> direction” in revision. For the crystal facet, the format of (101) is preference. For the crystal direction, the format of <101> is a standard style.

5) Line 81 “suit” should be “sun”.

Answer: Thanks very much. We have made it different in revision.

6) Line 103, 111, 112 and 161 “(101)” should be “[101]”.

Answer: Thanks very much. We have corrected it in revision.

7) Please pay attention to some small mistakes. For example, in the caption of Figure S1, “(a, d, and g)” has been used three times. The numbering of Figure S2-S4 is confusing.

Answer: Thank you very much. We have fully and carefully revised our manuscript.

Reviewer #3 (Remarks to the Author):

This manuscript reports on the growing and characterization of porous single-crystalline anatase Ti_nO_{2n-1} ($n = 7-38$) large size crystals and their application for photoelectrochemical oxidation of

benzene to phenol which showed good benzene conversion (60.1%) and excellent phenol selectivity (99.6%). The material was well characterized by variety of instrumentations and the manuscript is generally well written and clearly presented. Therefore, I recommend the manuscript to be published after minor revisions based on comments below:

Answer: Thank you very much. We have further conducted supplementary experiments to support the photochemistry mechanism. We have fully and carefully revised our manuscript and we sincerely wish these could well fit your requirements.

1- It is very difficult to understand the higher selectivity towards phenol formation since phenol is more reactive than benzene and the catalytic test was carried out for 24 h. Moreover, it is well known that the photo-generated OH radical generally shows poor selectivity. Therefore, reasonable explanations should be given for the high selectivity of P-SC Ti_nO_{2n-1} crystal toward phenol formation.

Answer: Thank you very much. The selectivity of photocatalytic oxidation of benzene to phenol can reach >99% in many reports; however, the conversion ratio is normally below 20%. In this photoelectrochemical oxidation process, a suitable amount of CH_3CN is necessary and it is used as solvent that increases the concentration of benzene in aqueous phase toward hydroxylation reactions while produced phenol would be extracted into the organic phase to fully minimize the overoxidation. We then summarize the conversion ratio of benzene to phenol in reported work as shown in **Table 1**. High selectivity of phenol is achieved with specific solvent in the photochemical oxidation process.

Table 1. Photocatalytic benzene hydroxylation to form phenol over different catalysts.

Catalyst	Photocatalysis	Photoelectrochemistry	Solvent	Duration	Conversion	Selectivity
MIL-100(Fe)	√		CH_3CN	24 h	20.1%	>98% ^[1]
Fe-g- C_3N_4 /SBA-15	√		CH_3CN	4 h	11.9%	94.3% ^[2]
Fe-CN/TS-1	√		CH_3CN	4 h	10.4	96% ^[3]
1%Au/ TiO_2	√		CH_3CN	4 h	3%	>99% ^[4]
Ti_9O_{17}		√	CH_3CN	24 h	60.1%	>99% ^[This work]

[1] Wang D.; Wang M.; Li Z. Fe-Based metal-organic frameworks for highly selective photocatalytic benzene hydroxylation to phenol. *ACS Catal.* **5**, 6852–6857 (2015).

[2] Chen, X.; Zhang, J.; Fu, X.; Antonietti, M.; Wang, X. Fe-g- C_3N_4 -catalyzed oxidation of benzene to phenol using hydrogen peroxide and visible light. *J. Am. Chem. Soc.* **131**, 11658–11659 (2009).

[3] Ye, X.; Cui, Y.; Qiu, X.; Wang, X. Selective oxidation of benzene to phenol by Fe-CN/TS-1 catalysts under visible light irradiation. *Appl. Catal. B* **152**, 383–389 (2014).

[4] Devaraji, P.; Sathu, N.; Gopinath, C. Ambient oxidation of benzene to phenol by photocatalysis on Au/Ti_{0.98}V_{0.02}O₂: role of holes. *ACS Catal.* **4**, 2844–2853 (2014).

2- The authors mentioned that the photo-generated OH radical is responsible for benzene oxidation (without any evidence). However, reasonable pathway is not clear i.e. is it through direct oxidation of H₂O and formation of OH radical or indirectly via in situ formation of H₂O₂?

Answer: Thank you very much. The benzene hydroxylation to produce phenol in photochemistry oxidation process is generally believed to proceed via an oxygenation pathway induced by the in situ-formed $\cdot\text{OH}$ radical. These active $\cdot\text{OH}$ radicals would readily oxidize the benzene to phenol in aqueous phase. We have further conducted electron spin resonance (ESR) measurement to detect the irradiated reaction system containing 5,5-dimethyl-1-pyrroline *N*-oxide (DMPO). Here the DMPO acts as a trapping agent which shows four typical signals for the DMPO- $\cdot\text{OH}$ adduct. As shown in Supplementary Figure 18, the observed ESR signals confirm the formation of $\cdot\text{OH}$ radicals during the photoelectrochemical reactions. And phenol is formed in this process. We then add ethanol, which is a kind of scavenger of $\cdot\text{OH}$ radicals, into the reaction system and find that negligible phenol is formed. Therefore, we could confirm that the reasonable pathway of benzene oxidation is a photochemistry process through a $\cdot\text{OH}$ radical reaction.

Supplementary Figure 18. DMPO spin-trapping ESR spectra for the $\cdot\text{OH}$ radical in the presence of porous single-crystalline Ti₉O₁₇ under different light intensities. (a). $10 \times \text{AM } 1.5\text{G}$ irradiation and (b). $1 \times \text{AM } 1.5\text{G}$ irradiation.

3- Additionally, phenol formation could be also initiated by photoinduced electron transfer from benzene and formation of benzene radical cation, which reacts with H₂O to yield OH - benzene adduct radical. Therefore, more detailed knowledge about the mechanism of photoelectrochemical benzene hydroxylation to phenol is required.

Answer: Thank you very much. We have further conducted supplementary experiment to detect the formation of benzene radical cation in photoinduced process. In this process, the aqueous solution after 1 hour reaction during the photoelectrochemical test is immediately cooled to solid phase using liquid nitrogen. We immediately detect the typical signal of electron spin resonance (ESR) of benzene radical cation. In our work, we have not observed the corresponding ESR signal, which confirms the absence of benzene radical cation during the photoelectrochemical reaction.

Therefore, the reasonable pathway of benzene oxidation is a $\cdot\text{OH}$ radical reaction.

4- The high performance of P-SC $\text{Ti}_n\text{O}_{2n-1}$ is attributed to the presence of Ti(III) interstitials which enhance visible-infrared light absorption. Why then the catalytic activity of P-SC Ti_7O_{13} is less than P-SC Ti_9O_{17} ?

Answer: Thank you very much. Yes, the high performance is attributed to the visible-infrared light absorption through band gap engineering by control of Ti interstitials in lattice. The porous single crystalline Ti_9O_{17} shows better photoelectrochemical performance than porous single crystalline Ti_7O_{13} under the identical conditions even though the Ti_7O_{13} demonstrates the further enhanced visible-infrared light absorption. It should be noted that the transport properties of porous single crystals would play another role for the enhancement of electrode activity. As shown in **Figure 5d** and **Supplementary Figure 14**, the porous $\text{Ti}_n\text{O}_{2n-1}$ single crystals show enhanced transport properties with the decrease of n values; however, the transport property of the porous Ti_7O_{13} single crystal is inferior to that of porous Ti_9O_{17} single crystal, which could be due to the heavy defect concentration of Ti interstitials in lattice in the porous Ti_7O_{13} single crystal.

Supplementary Figure 14. (a) The resistivity, carrier density, Hall coefficient and Hall mobility of $\text{Ti}_n\text{O}_{2n-1}$ single crystals growth along the b -axis of the KTP substrates. (b) The resistivity, carrier density, Hall coefficient and Hall mobility of $\text{Ti}_n\text{O}_{2n-1}$ single crystals growth along the c -axis of the KTP substrates.

5- In Figure 6a, the authors mentioned that the Photoelectrochemical reactions were carried out in NaOH while in the manuscript (page 4) as well as in Figure 8 (SI) in KOH!

Answer: Thank you very much. We have corrected them in revision.

6- Please delete one of from Figure 9 “(b) Durability test of of”

Answer: Thank you very much. We have corrected them in revision.

REVIEWERS' COMMENTS:

Reviewer #1 (Remarks to the Author):

The authors added some more experimental data to clarify the results. But they should add more explanations on the excuse that (101) crystal facet should be focused even with lower surface energy. Is it because of synthesis problem? And also thickness of TiO_2 is 0.5 μm , it is so thick to have enough mobility of electron/hole due to limited diffusivity of Ti-O photocatalysts such as 200 μm . They should add more excuse of it.

Reviewer #2 (Remarks to the Author):

The authors have well addressed my requested revisions. They have added adequate data and sufficient discussion related to electrode microstructure, Raman spectroscopies and applied bias photo-to-current efficiencies. The experimental details about photoelectrochemical measurement are added in revision. The errors about facet expression and figure captions are well corrected. I recommend to accept this manuscript in its current form.

Reviewer #3 (Remarks to the Author):

The authors have responded to all my concerns and questions and made the necessary changes to the manuscript. Therefore, I support publication of the manuscript in the present form.

Response to reviewers

Reviewers' comments:

Reviewer #1 (Remarks to the Author):

The authors added some more experimental data to clarify the results. But they should add more explanations on the excuse that (101) crystal facet should be focused even with lower surface energy. Is it because of synthesis problem? And also thickness of $\text{Ti}_n\text{O}_{2n-1}$ is 0.5 mm, it is so thick to have enough mobility of electron/hole due to limited diffusivity of Ti-O photocatalysts such as 200 nm. They should add more excuse of it.

Answer: Thank you very much for your kind comments. The (101) facet dominates the growth of porous single-crystalline (P-SC) $\text{Ti}_n\text{O}_{2n-1}$ (n=7-38) in our work, which may be attributed to that the (101) facet has the lower surface free energy among the different crystal facets in anatase TiO_2 . In this work, we therefore focus on the growth of P-SC $\text{Ti}_n\text{O}_{2n-1}$ (101) crystals which are bulk porous and we then investigate their properties and photoelectrochemical performance. Our future work will be focused on other crystal facets with high surface free energy and these facets with special orientations could be obtained from large-size P-SC TiO_2 single crystals by crystal processing and cutting technologies.

We fully agree with the referee on the short diffusion length of charge carrier in many photocatalysts including TiO_2 oxide, which could be also the reason why the thickness is ranging from several hundred nanometers to several micrometers for the poly-crystalline TiO_2 film photoanodes. The thickness is 0.5 mm for all the P-SC $\text{Ti}_n\text{O}_{2n-1}$ (n=7-38) single crystals in our work. These porous single crystals have excellent transport properties and high conductivities can be obtained with small n values. The structural coherence delivers high electron mobility in the P-SC $\text{Ti}_n\text{O}_{2n-1}$ (n=7-38) single crystals according to our Hall tests.

In our work, the macro-scale P-SC $\text{Ti}_n\text{O}_{2n-1}$ (n=7-38) single crystals (10 mm × 10 mm × 0.5 mm) are bulk porous with three-dimensional open frameworks. The diameter of a single $\text{Ti}_n\text{O}_{2n-1}$ skeleton is around 100-200 nm, which well fits the diffusion length of charge carrier and thus maximizes the photoelectrochemical conversion. The porous microstructure in $\text{Ti}_n\text{O}_{2n-1}$ single crystals not only reduces the light scattering but also provide enough surface areas to accommodate the surface reactions in photochemical processes.

Reviewer #2 (Remarks to the Author):

The authors have well addressed my requested revisions. They have added adequate data and sufficient discussion related to electrode microstructure, Raman spectroscopies and applied bias photo-to-current efficiencies. The experimental details about photoelectrochemical measurement are added in revision. The errors about facet expression and figure captions are well corrected. I recommend to accept this manuscript in its current form.

Answer: Thank you very much for your kind recommendation.

Reviewer #3 (Remarks to the Author):

The authors have responded to all my concerns and questions and made the necessary changes to the manuscript. Therefore, I support publication of the manuscript in the present form.

Answer: Thank you very much for your kind suggestion.